# Fine-Tuning BERT Models for Intent Recognition Using a Frequency Cut-Off Strategy for Domain-Specific Vocabulary Extension

Fernando Fernández-Martínez [1,*], Cristina Luna-Jiménez [1], Ricardo Kleinlein [1], David Griol [2], Zoraida Callejas [2] and Juan Manuel Montero [1]

1    Grupo de Tecnología del Habla y Aprendizaje Automático, Information Processing and Telecommunications Center, E.T.S.I. de Telecomunicación, Universidad Politécnica de Madrid, Avda. Complutense 30, 28040 Madrid, Spain; cristina.lunaj@upm.es (C.L.-J.); ricardo.kleinlein@upm.es (R.K.); juanmanuel.montero@upm.es (J.M.M.)
2    Department Software Engineering, University of Granada, CITIC-UGR, Periodista Daniel Saucedo Aranda S/N, 18071 Granada, Spain; dgriol@ugr.es (D.G.); zoraida@ugr.es (Z.C.)
*    Correspondence: fernando.fernandezm@upm.es

**Abstract:** Intent recognition is a key component of any task-oriented conversational system. The intent recognizer can be used first to classify the user's utterance into one of several predefined classes (intents) that help to understand the user's current goal. Then, the most adequate response can be provided accordingly. Intent recognizers also often appear as a form of joint models for performing the natural language understanding and dialog management tasks together as a single process, thus simplifying the set of problems that a conversational system must solve. This happens to be especially true for frequently asked question (FAQ) conversational systems. In this work, we first present an exploratory analysis in which different deep learning (DL) models for intent detection and classification were evaluated. In particular, we experimentally compare and analyze conventional recurrent neural networks (RNN) and state-of-the-art transformer models. Our experiments confirmed that best performance is achieved by using transformers. Specifically, best performance was achieved by fine-tuning the so-called BETO model (a Spanish pretrained bidirectional encoder representations from transformers (BERT) model from the Universidad de Chile) in our intent detection task. Then, as the main contribution of the paper, we analyze the effect of inserting unseen domain words to extend the vocabulary of the model as part of the fine-tuning or domain-adaptation process. Particularly, a very simple word frequency cut-off strategy is experimentally shown to be a suitable method for driving the vocabulary learning decisions over unseen words. The results of our analysis show that the proposed method helps to effectively extend the original vocabulary of the pretrained models. We validated our approach with a selection of the corpus acquired with the Hispabot-Covid19 system obtaining satisfactory results.

**Keywords:** topic classification; intent detection; conversational systems; recurrent networks; attentive RNN; attentive LSTM; transformer models; transfer learning

## 1. Introduction

Spoken language understanding (SLU) in conversational systems is traditionally divided into two main subtasks: intent detection and semantic slot filling, both extended with domain recognition for multidomain dialogue systems [1,2].

Intent detection or recognition (sometimes also called intent classification) is the task of classifying user utterances into previously defined intent categories. This classification is based on what the user wants to achieve [3,4]. These users' intents can also be denoted as dialogue acts, which are often defined as update operations on information states (or contexts). These state-changing operations or actions are produced and updated by the

users share in the dialogue to fulfill their needs, wants, and purposes (e.g., ask for train schedules, book a hotel, etc.).

Intent classification is a complex process that normally involves several key challenges. First, adequate data sources to apply statistical methodologies are often not available (very few public corpora have intent annotations, which indeed may be difficult and time-consuming to obtain) [5]. Another important challenge is the irregularity of users' expressions. Users tend to use colloquial expressions and short sentences, which may cause identifying user's intents to be very difficult [6]. Finally, the use of broad content, implicit intents (i.e., when the user does not have clear intent requirements, and therefore the user's intent must be inferred from the set of possible intents defined for the task), or multiple intents (i.e., when the user refers to more than one intent) are other examples of challenging situations that cause intent detection to be particularly difficult [7,8].

Intent classification can be particularly useful for conversational systems to empower customer services with AI-driven FAQ software [1]. Its usefulness in this domain is twofold: first, the intent recognizer may help detecting the main information pieces that are present in the user's utterances, thus becoming a solution for completing the natural language understanding task; then, similarly, the dialogue management task could also be accomplished by simply assigning the user utterance to one of the intents defined, the one corresponding to the adequate response from the system.

In this paper we present an exploratory analysis in which different deep learning (DL) models for intent detection and classification were evaluated. Particularly, we compared and analyzed traditional RNN and transformer models, an emergent family of models that achieve state-of-the-art performance in numerous natural language processing applications. Our first approach is based on the typical recurrent neural network model with word embeddings as its first layer. The results of the evaluation of this model shows an improvement of performance when using named-entities for intent recognition (as an alternative to using raw text with basic preprocessing) and embeddings adapted to the specific task (compared to using "out-of-the-box" embeddings trained on general text corpora).

Our second approach is based on a transformer model that was developed by fine-tuning a BETO model (a Spanish pretrained bidirectional encoder representations from transformers (BERT) model from the Universidad de Chile) for the intent detection task. Our experiments in this case confirmed that best performance was achieved using transformer models.

In addition to the comparison between models, we also analyze the effect of inserting unseen domain words to extend the vocabulary of the model as part of the fine-tuning or domain-adaptation process. In this regard, transformers can be quite inefficient in learning new domain words, particularly those that are not backed up with sufficient domain-specific training data. This suggests that it might be preferable to not include every unseen word but just those that are best for bootstrapping their embeddings from the available data. Hence, a very simple word frequency cut-off strategy is proposed and validated as a suitable method for vocabulary addition. The results of our analysis show that the proposed method may help to effectively extend the original vocabulary of the pretrained models, by specifically including only those unseen words with the highest occurrence frequency in the domain-specific training data. The lessons learned from our experiments and experience were reported in the next sections.

All the suggested approaches were validated with a selection of the corpus acquired with the Hispabot-COVID-19 conversational system, which was developed by the Spanish government to provide responses to FAQ related to the pandemics originated by the COVID-19.

## 2. Literature Review

As it was described in the previous section, user intent detection plays a critical role in question-answering and dialogue systems. Traditional intent detection methods include rule-based template semantic recognition methods and methods based on the use

of statistical features, such as Naive Bayes, AdaBoost, support-vector machines, and logistic regression [3,4].

Current mainstream methods are mainly based on DL techniques and the use of word embeddings, which were probed as a solution to better representational ability and domain extensibility instead of using bag of words [9].

Intent detection methods based on DL techniques can be classified into methods using convolution neural networks [10], recurrent neural networks [11] and their variants (LSTMs and GRUs) [4,7], the Bidirectional Long short-term Memory (BLSTM) self-attention model [12], the capsule network model [13], the method of joint recognition [14], the use of distances to measure the text similarities (such as TF-IDF) [15], or methods combining several DL models [16,17]. Nonetheless, the appearance of BERT (Bidirectional Encoder Representations from Transformers) models greatly contributed to enhance natural language processing [18]. BERT-like models are based on the most commonly used transformer architecture [19] for representation learning. These models proved to be extremely flexible, demonstrating that a single pretrained BERT model can be successfully fine-tuned to achieve state-of-the-art performance on a wide variety of NLP applications [20,21].

*Domain-Specific BERT Models*

Different works already showed that BERT models' performances on many natural language processing tasks, including intent recognition, drop dramatically on held-out data when a significant percentage of out-of-vocabulary (OOV) words do not appear in the training data [22,23].

For instance, in [24], Wang et al. investigated two different approaches aimed at enlarging the vocabulary size when fine-tuning a pretrained multilingual BERT model on a variety of NLP tasks, thus addressing the OOV problem: joint mapping and mixture mapping. Their experimental results show that using mixture mapping is more promising.

Alternatively, Poerner et al. suggested a different approach for domain adaptation [25]. Here, authors propose to train Word2Vec on target-domain text and align the resulting word vectors with the wordpiece vectors of a general-domain pretrained language model (PTLM). Authors claim their method is cheaper compared to that of successful but expensive unsupervised pretraining on target-domain text. Besides testing their approach on several biomedical named entity recognition (NER) tasks, they also proved their method to be effective when adapting an existing general-domain question answering (QA) model to an emerging domain: the COVID-19 pandemic.

Similarly, the work authored by Carrino et al. [26] presents different biomedical and clinical transformer-based pretrained language models for Spanish and analyzes the impact of the model's vocabulary on the NER performances. Authors experimented with different pretraining choices, such as masking at word and subword level, varying the vocabulary size and testing with domain data, looking for better language representations. Mixed-domain pretraining and cross-domain transfer approaches were confirmed as a valid alternative to generate suitable target models in the absence of enough data to train a model from scratch. Their results also confirmed that domain-specific pretraining is fundamental to achieving higher performances in downstream NER tasks, even within a midresource scenario.

Tai et al. recently introduced exBERT [27], a training method to extend BERT pretrained models from a general domain to a specific domain with a new additive vocabulary under constrained training resources (i.e., constrained computation and data). exBERT uses a small extension module to learn to adapt an augmenting embedding for the new domain in the context of the original BERT's embedding of a general vocabulary. Authors demonstrate that exBERT consistently outperforms prior approaches when using limited corpus and pretraining computation resources.

In [28], Koto et al. evaluate different methods for extending an Indonesian BERT model with additive domain-specific vocabulary. They specifically focused on efficient model adaptation under vocabulary mismatch and benchmarked different ways of initializing the

BERT embedding layer for new word types. The work suggests that initializing with the average BERT subword embedding may be more effective than other initialization methods and help to make training five times faster.

Finally, although recently proposed methods such as vocabulary expansion [25], vocabulary expansion, and continual pretraining [27] overcome the issue of OOV words, these methods also increase the size of vocabulary due to the addition of new terms in the vocabulary. In this regard, Yao et al. recently proposed the Adapt-and-Distill approach [29] to adapt general models to a specific domain using vocabulary expansion and knowledge distillation. Different from existing adaptation methods, this approach not only adapts general models to specific domain but also reduces the size of the model.

### 3. The Hispabot-COVID-19 Dataset

With the aim of providing information to citizens and facilitating contact with public institutions, the Spanish government launched in March 2020 Hispabot-Covid19, a conversational assistant [30] developed to answer frequently asked questions related to the pandemic originated by COVID-19 and its implications in Spain. The system received more than 350,000 queries between April and June 2020. The Hispabot-Covid19 assistant accessed official sources, such as the Spanish Ministry of Health and the World Health Organization, to provide information. The assistant also incorporated the information published in the Spanish BOE (official Spanish state bulletin) regarding the application of the State of Alarm and the Plan for the Transition towards a New Normality. The assistant did not require or analyze personal data.

A total of 164 intents were defined to report on various issues, such as: attention telephone numbers, symptoms, vulnerable groups, transmission, prevention, coexistence with infected people, and conditions for quarantine and isolation, among many others. Users' utterances were classified to provide a response associated to each one of them.

All dialogues were conveniently recorded in the system. With the aim of continuously retraining and improving the assistant, new training phrases were processed and new question categories were incorporated daily. Besides, misunderstandings were detected and corrected, and the knowledge base was accordingly restructured with the information updates provided. The proposal we present in this paper was validated upon a subset of the official Hispabot-Covid19 dataset, whose main features were summarized in Table 1.

**Table 1.** Main features of selection of Hispabot-Covid19 corpus used in paper.

| Param | Value |
|---|---|
| Number of samples | 7532 |
| Number of intents | 164 |
| Number of entities | 55 |
| Different words | 7658 |
| Average number of words per utterance | 6.16 |
| Average number of entities per utterance | 0.6 |
| Average number of words per entity | 1.22 |

In addition, we also present a histogram representation in Figure 1 with the amount of samples available per each category (in blue). Specifically, the name of the most relevant categories (i.e., the most frequent ones, those with 100 or more samples) were highlighted in bold above the corresponding line. Information about the average sample length per class is also provided (in orange). Mean values are also displayed (in dotted lines) for both data series.

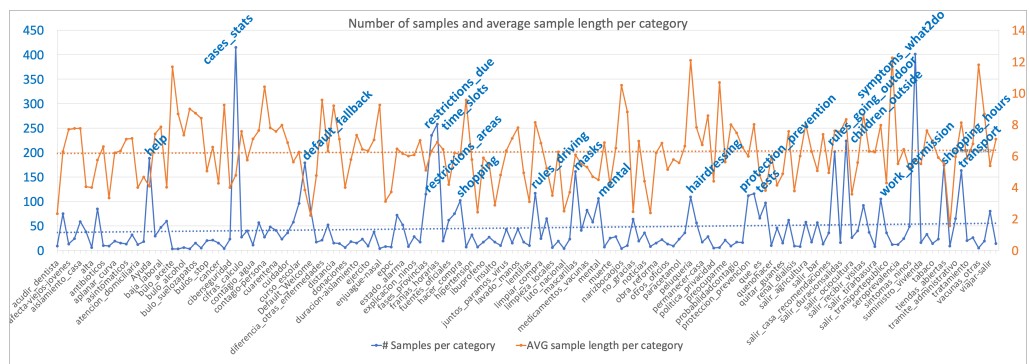

**Figure 1.** Histogram of amount of samples per class and average sample length per class.

## 4. Recurrent Neural Networks (RNN)-Based Model

Recurrent Neural Networks (RNN) were widely used in numerous NLP tasks, such as the analysis of Twitter messages, for example. In this paper, we will adopt them as our first solution for the intent classification task. When applied to text inputs, RNN models have the ability to process these inputs sequentially, performing the same operation, $h_t = f_W(x_t, h_{t-1})$, on each of the different words that constitute the text input sequence (i.e., their corresponding word embeddings, to be more exact), where $h_t$ is the hidden state, $t$ the time step, and $W$ the weights of the network.

Since this operation is formulated in such a way that the hidden state at each time step depends on the previous hidden states, the order of the words in our text sequences becomes particularly important. Consequently, RNNs allow us to handle input sentences of variable length, which turned out to be an essential aspect given the nature of our problem.

There are many different architectures for RNNs. Among them, we opted for the so-called long short-term memory (LSTM) networks [31]. LSTM networks are a kind of RNN that introduce a gating mechanism that ensures proper gradient flow through the network, thus helping to prevent the vanishing (and exploding) gradient problem that typically exists with standard RNNs. One of the most interesting features of LSTMs is that they are capable of learning long-term dependencies. Specifically, a common LSTM unit is basically composed of a cell, an input gate, an output gate, and a forget gate. These gates allow the network to control and update for each cell its corresponding state by deciding where information in a sequence of data is placed and that it is preserved from that provided by the previous cells, as well as what information is provided to the next ones.

### 4.1. Embeddings

(Word) embeddings are representations obtained for words used for text analysis, typically in the form of a real valued vector that encodes the meaning of the word, such that the words that are closer in the vector space are expected to be similar in meaning. In this paper, our RNN model makes use of fastText word embeddings [32]. The network receives a sequence of vectors (i.e., word embeddings) as input, which was obtained for each input sentence using the fastText tool. The fastText tool was recently open-sourced by Facebook Research, and it enables a fast and effective method to learn word embeddings. This tool proved to be effective and useful in different applications such as text classification, clustering, and information retrieval.

FastText is similar to the Word2Vec approach [33] although it builds on Word2Vec's specific limitation. Specifically, FastText can handle new, OOV terms by extending the Word2vec skip-gram (SG) model with internal subword information in the form of character n-grams (i.e., sequences of adjacent characters). The method can build a vector representation for a word based on its constituent subparts (or subword components), which allows the model to represent the morphology and lexical similarity of words, in addition to being able to construct vectors for unseen words. This feature is particularly helpful to enhance learning on heavily inflected languages [34]. This would trigger, for example, the words

close, closer, and closed to all have similar embeddings/vectors, even if they tend to appear in different contexts.

### 4.2. Model Description

Our approach is based on a two-layer, bidirectional LSTM model with a deep self-attention mechanism that is represented in Figure 2. The model is implemented in Pytorch [35] and based on the architecture proposed in [36].

#### 4.2.1. Embedding Layer

The model is designed to work with sequences of words as inputs, thus allowing us to process any type of sentence. For this, a first embedding layer is provided that collects the embeddings $\vec{x}_1, \vec{x}_2, \ldots, \vec{x}_N$ corresponding to each of the words $w_1, w_2, \ldots, w_N$ constituting the sentence we want to process, where $N$ is the number of words in our sentence. We initialize the weights of the embedding layer with our pretrained word embeddings.

#### 4.2.2. Bi-LSTM Layer

Unlike conventional standard LSTM models, which behave in a unidirectional way, *bidirectional LSTM* (Bi-LSTM) models allow us to collect such information in both directions.

A standard LSTM network takes as input the direct sequence of word embeddings and produces the outputs $\vec{h}_1, \vec{h}_2, \ldots, \vec{h}_N$, where $\vec{h}_i$ is the hidden state of the LSTM cell at time step $i$, summarizing all the information that the network accumulated from our sentence up to word $w_i$. Conversely, a Bi-LSTM consists of two LSTMs, a *forward* $\overrightarrow{LSTM}$ that allows the analysis of the sentence from $w_1$ to $w_N$ and an *inverse or backward* $\overleftarrow{LSTM}$ that allows a similar analysis to be carried out but in the opposite direction, from $w_N$ to $w_1$. The outputs of the Bi-LSTM layer are then obtained by simply concatenating for each word the outputs obtained from the analysis performed in each specific direction (as it can be deduced from Equation (1) where $||$ corresponds to the concatenation operator and $L$ to the size of each LSTM).

$$h_i = \overrightarrow{h_i} || \overleftarrow{h_i} \text{ , where } h_i \in R^{2L} \tag{1}$$

$$e_i = g(h_i) \tag{2}$$

$$a_i = exp(e_i) / \sum_{j=1}^{N} exp(e_j) \tag{3}$$

#### 4.2.3. Attention Layer

Our model is able to identify top most informative words by means of a deep self-attention mechanism. Thus, actual importance and contribution of each word is estimated as a non-normalized attention score (Equation (2)) by means of a multilayer perceptron (MLP) composed of two layers with a nonlinear activation function (*tanh*) similar to that proposed in [37]. Then, the attention scores are normalized as in Equation (3).

The attention function $g$ is learnt by the MLP as a probability distribution on the hidden states $h_i$, that allows us to obtain the attention coefficients $a_i$ that each word receives. The weights of the attention layer encoding such function $g$ are learned together with the rest of the model parameters using backpropagation. The output of the attention layer is simply computed as the convex combination $r$ of the contextual embeddings (i.e., the Bi-LSTM outputs $h_i$) with weights $a_i$ (a convex combination is a linear combination where all the coefficients are non-negative and add up to one).

#### 4.2.4. Output Layer

Finally, a task-specific, fully-connected layer performs the classification function based on the above mentioned feature vector $r$. This layer is followed by a *softmax* operation, which outputs the probability distribution over the classes.

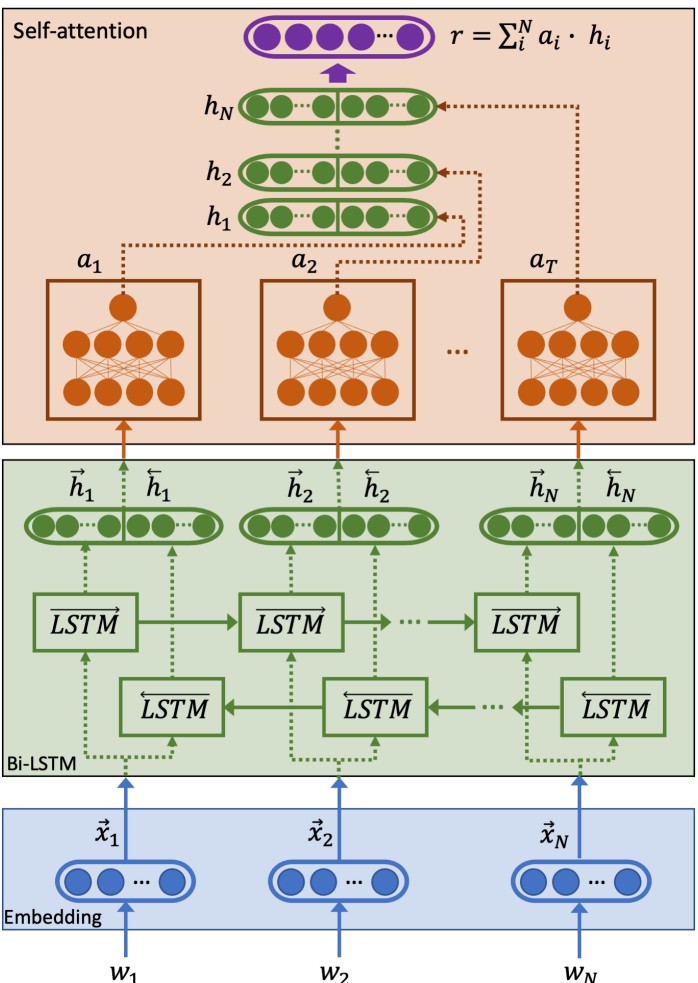

**Figure 2.** Proposed Recurrent Neural Network (RNN) model (figure directly taken from [38]).

## 5. Transformer Based Model

Transformer based models are a natural choice when working with text data. Like recurrent neural networks (RNNs), transformers are also designed to handle sequential input data, such as text sentences. However, unlike RNNs, transformers do not necessarily process the data in order. Transformers overcome this limitation by applying self-attention to compute in parallel for every word in a sentence or document an "attention score" to model the influence each word has on another [39]. Due to this feature, transformers allow for greater parallelization and thus can be trained faster than RNNs. This makes it possible to efficiently train very big models on large amounts of data that we can later take advantage of by fine-tuning them on down-stream tasks.

For this task we decided to use BERT. These models usually receive sentences as inputs that are divided into single tokens, obtaining a sequence of them. BERT models use context from both directions. Specifically, the transformer processes any given word in relation to all other words in a sentence, rather than processing them one at a time. Besides, BERT's self-attention mechanism helps to identify what the keywords in a sentence are.

The way this process is carried out depends on the tokenizer used. BERT's tokenizer is based on words and subwords so, for instance, if a word is not included in the original vocabulary, it will be divided in a sequence of subtokens that all together would form the initial word. Unfortunately, when fine-tuning our model to a specific task or domain, this happens frequently with many common words related to the topic. Hence, what we decided to test is whether adding new tokens to our initial vocabulary may result in a better intent recognition. To select what new tokens should be added, we specifically target

those most frequent words in our training dataset that were not already included in the original tokenizer vocabulary.

*Model Adaptation*

Pretrained DL models (including transformers) are increasingly used on an increasing variety of tasks, including NLP. Because they are trained on large corpora (see, e.g., BERTlarge [18] or RoBERTa [40]), pretrained models can learn universal language representations, which can be beneficial for downstream NLP tasks [41], such as intent recognition. Pretrained NLP models allow to avoid the need for training a new model from scratch, thus requiring less training data and less training time compared to that of custom-built models, and they often lead to breakthrough performance improvements. However, although the adoption of reusable pretrained NLP models help developers to quickly build NLP applications, there are some important challenges that still need to be overcome [27]. Specifically, conventional fine-tuning methods still often fail when extending pretrained models with domain-specific vocabulary, thus leading to suboptimal performance on downstream tasks. In some cases, such as with the Hispabot-Covid19 domain on which this paper focuses, it may be crucial to enrich the vocabulary of the pretrained model with that from our specialized domain. Sentences in the target domain may combine words from both the original language model's vocabulary and new domain-specific vocabulary. Thus, designing new methods able to effectively handle this mixture of vocabulary becomes essential in successfully adapting the pretrained model to the new domain [42].

In this work we suggest the use of a well-known method to address this issue explicitly. Aside from reusing and adapting the original vocabulary, our method proposes to extend it by adding only a subset of the top most relevant words from the new domain's vocabulary that were not already included in the original one. This reduced subset of words is carefully selected in a rational way by simply but effectively attending to their actual frequency in our training data. Specifically, this method may help in achieving an adequate trade-off between performance improvement, which may be significantly enhanced, and the need for computational resources and training data, which may be both significantly reduced.

## 6. Evaluation

In this section, we detail the experimental setup adopted for evaluation purposes and compare and discuss the performance of the proposed approaches.

### 6.1. Basic Preprocessing of Training Data

A simple preprocessing is first applied to all the sentences in our dataset. We first preprocess the input text to remove some special characters and transform it into a list of words. Words are then converted into lowercase. Next, special words including emails, percentages, money, phone numbers, times, dates, URLs and/or hashtags are assimilated and replaced by special tokens, such as MAIL, DATE, URL, etc., to prevent information from being lost during data representation.

### 6.2. Named Entity Recognition (NER)

Named entity recognition (NER) (also known as entity identification, entity chunking, and entity extraction) is a subtask of information extraction that seeks to locate and classify elements in text into predefined categories, such as the names of persons, organizations, locations, expressions of times, quantities, monetary values, percentages, etc. [43]. In this work, we did not implement a named entity recognizer and simply relied on manually annotated entities. All the sequences of words in the sentences of the dataset that are linked to relevant entities for the intent recognition task, such as dates, activities, symptoms, or COVID-19 and location names, were carefully identified and annotated. Hence, the dataset comes with full annotation of entity-related items that will be useful for engineering feature extractors for named entity recognition in the future.

### 6.3. General Experimental Setup

To prevent overfitting, all the experiments were carried out following a five-fold cross-validation scheme. Each model was trained for 100 epochs using 'Stop-Early' as the stopping criterion. No exhaustive exploration of the hyperparameters of our models was conducted. Models were trained using an Adam optimizer [44], with initial learning rate of 0.001, batch size of 32, and early-stopping after five epochs without improvement in the F1 classification score. For the calculation of F1, we used the *weighted* version that takes into account the number of examples available for each different class. Confidence intervals were estimated to evaluate the significance of our methods and their proper comparison.

#### 6.3.1. RNN Specific Setup

With regards to the RNN model both the Bi-LSTM and attention layers had a 0.3 dropout rate. The encoder layers had a size *L* of 200. Other values were also tested, though size 200 yielded best performance. As a way to increase input variability from epoch to epoch, white noise was randomly added to input embeddings with 0.15 probability rate to increase the robustness of the model. Figure 1 shows that some classes have more training examples than others. Hence, to prevent introducing bias in our models, we apply class weights to the loss function, penalizing more the misclassification of under-represented classes. These weights are computed as the inverse frequencies of the classes in the training set.

#### 6.3.2. BERT Specific Setup

Our BERT model was implemented and fine-tuned for the intent classification task using the simple transformers library [45]. Although pretrained tokenizers work at both word and subword levels, the top N new tokens to be added to the vocabulary (i.e., those that happen to be the most frequent in our training data) were included as word-level units. The rest of new tokens, those connected to more infrequent words, just get split into smaller units to ensure that there are no out-of-vocabulary tokens and all vocabulary units get updated reasonably frequently during training.

### 6.4. RNN Model Evaluation

When using the RNN model the words in the input sentence (or their corresponding entities) need to be mapped and transformed into their corresponding embeddings, as the network can only work with numeric representations of each word. In this regard, those words that are present in our dataset but not in the pretrained word vectors were considered OOV words and were simply discarded. The type of word embeddings used in this work is described in Section 6.4.1. Specifically, the two different sets of word embeddings that were applied are detailed there. Once the input text is transformed into word embeddings by the Embeddings layer, the network is used to finally classify the text according to the identified classes of the problem at hand.

#### 6.4.1. RNN Word Embeddings

When available training resources are significantly constrained or limited, most NLP pipelines typically use pretrained word embeddings as the inputs of newly developed machine learning models. Therefore, for our first approach we decided to use a set of pretrained fastText vectors made up of a total of 2 million of 300-dimensional embeddings. These embeddings were generated for the Spanish language from a massive amount of different texts automatically extracted from Wikipedia [46] and other online sources.

Alternatively, we also used fastText to train task-specific embeddings from scratch. A setup similar to the one used for general pretrained models was adopted. As a result, we also obtained CBOW models but with a smaller size: dimension was set to 100 in this case, consistently with the smaller size of our dataset.

### 6.4.2. RNN Results

We evaluated three different models whose main characteristics and results are detailed in Table 2 (where statistical significance is highlighted in bold). The first model, which was adopted as our baseline, was directly trained from the available sentences after applying them the basic preprocessing described in Section 6.1. General pretrained embeddings were used for the tokenized words.

Compared to our baseline, the second model successfully introduced the use of the named-entities version of our dataset, while the third one also successfully combined it together with task-specific embeddings trained on our dataset.

Finally, although omitted in this work, error analysis indicates that errors are mainly due to semantically overlapped categories, such as "shopping" and "opening hours" or "traveling" and "transports", which suggests that a simplified and reduced set of intent categories may be also considered.

### 6.5. BERT Model Evaluation

Table 3 allows to compare the performance of different pretrained BERT models after fine-tuning them on our downstream task. In our analysis, we focus on comparing two different strategies for fine-tuning the pretrained models:

- first group of results (models 4 to 6 in the table) correspond to conventional fine-tuning without vocabulary extension;
- second group of results (models 7 to 9 in the table) correspond to the alternative fine-tuning approach, which makes use of our vocabulary extension method. In this case, for simplicity, only the top-performing configuration was included (i.e., for the specific number of 50 new tokens).

Both strategies were evaluated for three different pretrained models:

1. *BERT-base-multilingual-cased*: standard BERT multilingual base model pretrained on the top 104 languages with the largest Wikipedia using a masked language modeling (MLM) objective [18]. This model is case sensitive: it makes a difference between "english" and "English".
2. *dccuchile/bert-base-spanish-wwm-cased*: cased version of BETO (i.e., case sensitive), a BERT model trained exclusively on a big Spanish corpus [47]. Better results were reported for different tasks when relying on this Spanish model compared to that of other BERT-based models pretrained on multilingual corpora.
3. *dccuchile/bert-base-spanish-wwm-uncased*: the uncased version of BETO.

**Table 2.** Summary of results for RNN model.

| Model | Inputs | Embedding Type | F1 (%) |
|---|---|---|---|
| 1 | Preprocessed | General pretrained | 67.72 |
| 2 | NER based | General pretrained | 72.58 |
| **3** | **NER based** | **Domain-specific** | **75.33** |

### 6.5.1. BERT Results

The table shows that even without explicitly adding any new domain-specific vocabulary, the most simple version of our BERT based model (i.e., model 4) achieves comparable performance to our top performing RNN-based model (i.e., model 3). Nonetheless, performance was demonstrated to be significantly better for the model fine-tuned from the cased version of BETO (i.e., model 5), which confirms the superiority of this BERT-based approach over the evaluated RNN models.

Then, besides comparing different pretrained models' performance when adopting a conventional fine-tuning strategy, we also measured the impact of our vocabulary extension method when adapting them to the downstream task. As also shown in Table 3 (where statistical significance is highlighted in bold), the adoption of the proposed method allows us to effectively extend the original vocabulary of the pretrained models, thus consistently

improving their performance. Besides, the cased version of BETO clearly outperformed again the other two models: the uncased BETO version and the standard BERT multilingual base model.

**Table 3.** Summary of results for bidirectional encoder representations from transformers (BERT) model (named entity recognition (NER)-based inputs in all cases).

| Model | Pretrained Model | Extension | F1 (%) |
|:---:|:---:|:---:|:---:|
| 4 | bert-base-multilingual-cased | | 75.27 |
| **5** | **dccuchile/bert-base-spanish-wwm-cased** | No | **78.16** |
| 6 | dccuchile/bert-base-spanish-wwm-uncased | | 76.85 |
| 7 | bert-base-multilingual-cased | | 77.64 |
| **8** | **dccuchile/bert-base-spanish-wwm-cased** | 50 | **79.48** |
| 9 | dccuchile/bert-base-spanish-wwm-uncased | | 77.50 |

6.5.2. Analyzing the Effect of the Amount of New Words to Be Included

A more thorough and complementary analysis of the performance of our vocabulary extension method can be derived from Figure 3 where results for different values of N were presented (i.e., the number of new tokens or words included as new words) for the three alternative approaches. First, baseline performances obtained when the three different pretrained models are simply fine-tuned without explicitly adding any new word were also included at the beginning of the series for easier comparison (i.e., first 0 columns on the left). Results are then reported starting from 25 new tokens at first and up to 100, increasing the amount by 25 on each different and independent run. Further results for experiments with values of 200, 300, 400, and 500, respectively, are also reported.

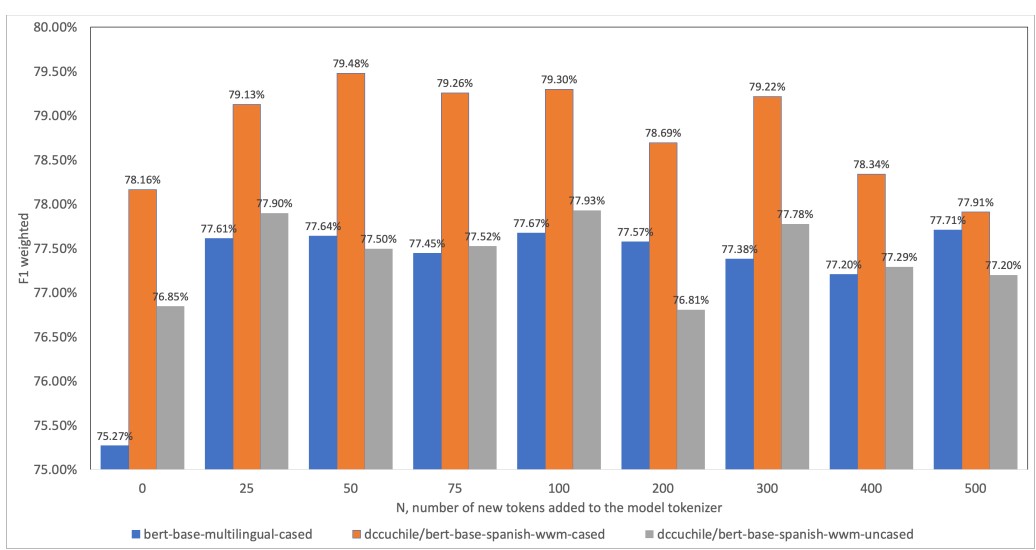

**Figure 3.** BERT models comparison when fine-tuning with our vocabulary extension strategy.

The figure shows that our vocabulary extension method consistently outperforms the prior approach based on general vocabulary. Particularly, our vocabulary extension method demonstrates to be effective achieving a top performance of 79.48% (achieved by the 50 new tokens configuration) and an improvement over the baseline performance obtained when our model is simply fine-tuned without explicitly adding any new word (i.e., 78.16%, as also previously reported in Table 3). However, results become worse when the amount of new tokens exceeds a certain small limit, which suggests the importance of finding an adequate balance between the increased complexity of our target model (i.e., the number of new unit embeddings to be learned) and the available training data. Finally, we can clearly confirm again that fine tuning the cased version of the BETO pretrained model is the top-performing approach when adopting our vocabulary extension method.

6.5.3. Analyzing the Effect of Different Data Pre-Processing Methods

Adopting the top-performing approach based on the cased version of the BETO pretrained model as a reference, we decided to further explore the use of some of the most popular text preprocessing techniques to find out whether they are actually useful or not. Evaluated techniques include the following:

- Stop word removal: we remove words conveying low-level information (such as articles, prepositions, pronouns, conjunctions, etc.) to give more focus to the important ones. The removal of stop words is expected to reduce the vocabulary size (i.e., model complexity), and thus to reduce the training time due to the fewer number of tokens involved in the training. We used *spaCy* library [48] to remove all the stop words.
- Punctuation removal: punctuation marks are often referred to as unhelpful or noisy parts of the text data that we should get rid of before training our models. Here, we also used the *spaCy* library to remove all of them.
- Lemmatization: lemmatization is another method to normalize the text (i.e., to keep the vocabulary small, which helps to improve the accuracy of many language modeling tasks). Particularly, lemmatization can be applied to generate the root form of the words. Again the *spaCy* library was used for this.
- Basic text normalization: special words including emojis, emails, percentages, money, phone numbers, times, dates, URLs and/or hashtags are assimilated and replaced by special tokens, such as MAIL, DATE, URL, etc., to prevent information from being lost during data representation.

The obtained results were summarized and sorted in terms of performance in Table 4. None of the applied techniques that are aimed at removing tokens were found to be effective. In short, DL methods that use embedding representations seem to not require any removing. Specifically, in that n-dimensional vector words like "dogs" and "dog" would already be closer to each other. So, the need to lemmatize becomes unnecessary.

**Table 4.** Results obtained for our top-performing approach based on cased version of BETO pretrained model when enriched with 50 new tokens and applying a specific text preprocessing technique (five-fold CV results). We used *weighted* version for all metrics to account for label imbalance (this can result in an F-score that is not between precision and recall).

| Technique | F1 | Precision | Recall |
|---|---|---|---|
| removing punctuation | 78.22 | 80.38 | 78.65 |
| without basic preprocessing | 77.75 | 80.10 | 78.01 |
| lemmatization | 76.91 | 79.37 | 77.14 |
| removing stopwords | 73.88 | 77.15 | 74.20 |

With regards to stop words and punctuation marks, we observe that our intent detection task cannot be accomplished properly after their removal. The removal of either stop words or punctuation marks is highly dependent on the task we are performing and the goal we want to achieve. In this case, we can confirm that the effect of removing them was clearly negative. Generally speaking, we should not remove anything (e.g., a word or a punctuation mark) that could be useful in some way.

Again, DL models working with vector embeddings, similarly to lemmatization, are currently the best methods to handle and filter those irrelevant terms. With regards to punctuation it is important to clarify that our word embeddings support punctuation and special symbols. Hence, it is reasonable to confirm that in this scenario it is better to retain punctuation as our model works better with it (removing the punctuation from the text is fine if your word embedding model does not support punctuation).

Finally, basic text normalization proved to be successful (i.e., performance decreases if we omit it). In this case, the process of transforming some words into their single canonical form still helps our model by reducing the number of unique words (i.e., reducing the vocabulary size helps reducing the model complexity and improving its performance).

## 7. Conclusions

The significance of this research lies in several aspects. First, we presented the intent recognition results obtained from an exploratory analysis in which different deep learning (DL) models were evaluated on the Hispabot-Covid19 dataset. Particularly, we compared traditional recurrent neural networks (RNN) and state-of-the-art transformer models including standard bidirectional encoder representations from transformers (BERT) [18] and its BETO variant [47]. Our experiments demonstrate that transformer models yield best performance. Specifically, this result was achieved by simply fine-tuning the BETO model for the intent detection task.

Second, we proposed and validated a simple but effective method for extending our pretrained BERT models with task- or domain-specific vocabulary. The method specifically accounts for term frequencies to rank and select specific words to be added at the word level. The performance of the extended model was significantly improved. This approach could be particularly attractive to ad-hoc and special purposes, or very specific domains with unique vocabularies where limited training data are available. In spite of the promising results and applications of our proposed method, a number of issues require further research. Particularly, additional research should be performed toward suggesting solutions for automatically finding or identifying the exact and optimal amount of new tokens to be added (i.e., the precise value that achieves a good balance between model complexity and available training data).

Third, when using transformer-like models and testing different text preprocessing methods, preserving the raw structure of the texts by not removing anything while simply performing a very basic text normalization helps to achieve a better performance. This result allowed a better understanding of the actual importance and convenience of applying (or not) those popular text preprocessing techniques when using DL and/or embedding representations compared to that of traditional approaches based on different foundations.

**Author Contributions:** Conceptualization, F.F.-M., C.L.-J., R.K., D.G., Z.C. and J.M.M.; data curation, F.F.-M., D.G. and Z.C.; formal analysis, F.F.-M., C.L.-J., R.K., D.G., Z.C. and J.M.M.; funding acquisition, F.F.-M., D.G., Z.C. and J.M.M.; investigation, F.F.-M., C.L.-J., R.K., D.G., Z.C. and J.M.M.; methodology, F.F.-M., C.L.-J., R.K., D.G., Z.C. and J.M.M.; project administration, F.F.-M., D.G., Z.C. and J.M.M.; resources, F.F.-M., D.G., Z.C. and J.M.M.; software, F.F.-M. and C.L.-J.; supervision, F.F.-M., D.G., Z.C. and J.M.M.; validation, F.F.-M. and C.L.-J.; visualization, F.F.-M., C.L.-J., R.K. and J.M.M.; writing—original draft, F.F.-M.; writing—review & editing, F.F.-M., C.L.-J., R.K., D.G., Z.C. and J.M.M. All authors have read and agreed to the published version of the manuscript.

**Funding:** The work leading to these results was supported by the Spanish Ministry of Science and Innovation through the *R&D&i* projects GOMINOLA (PID2020-118112RB-C21 and PID2020-118112RB-C22, funded by MCIN/AEI/10.13039/501100011033), CAVIAR (TEC2017-84593-C2-1-R, funded by MCIN/ AEI/10.13039/501100011033/FEDER "Una manera de hacer Europa"), and AMIC-PoC (PDC2021-120846-C42, funded by MCIN/AEI/10.13039/501100011033 and by "the European Union "NextGenerationEU/PRTR"). This research also received funding from the European Union's Horizon2020 research and innovation program under grant agreement N° 823907 (http://menhir-project.eu, accessed on 2 February 2022). Furthermore, R.K.'s research was supported by the Spanish Ministry of Education (FPI grant PRE2018-083225).

**Data Availability Statement:** Restrictions apply to the availability of these data. Dataset is available from the authors with the permission of the Spanish Secretary of State for Digitalization and Artificial Intelligence (SEDIA).

**Conflicts of Interest:** The authors declare no conflict of interest.

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
