# Peer review of "Fine-Tuning BERT Models for Intent Recognition Using a Frequency Cut-Off Strategy for Domain-Specific Vocabulary Extension"

_applsci, doi:10.3390/app12031610_

Round 1

Reviewer 1 Report

The paper is quite interesting, although a little 'obscure' (I am referring to a possible non-specialized audience), and is quite well written. 

  Nonetheless, the Literature Review is too short, if we refer only to section 2, and is, in a way, scattered all over the paper. 

  It would be better, clearer, and more useful to enhance section 2, perhaps giving it the simpler title of "Literature Review", and to make it more comprehensive, so that the readers can refer to all the works used and cited by the Authors and can have a detailed overview of the studies in the field (at least the most recent) - more studies could be added. 

  Sections 4 and 5 are a valuable part of the paper, and they are very good, therefore no major concerns or major comments about them. 

  It would be good, nonetheless, to have a dedicated section inherent in the Methodology itself, where all the process is explained in detail, step-by-step, with the aims of clarity and reproducibility. 

  Section 6 associates Results (partly) with Discussion, and is quite robust; it would be possible to make it a little stronger by enhancing further the analysis and by summarizing the findings, in order to provide all the readers with a sort of 'rubric' useful to recap 'what above'. 

  Section 7 would have to be expanded. It should 'mirror' the (quite effective) introduction and should stress on the significance of the paper it its field of studies and on how the Authors have been able to achieve their research goals (little summary about that, which would complete the article in itself). 

  The English language, as told, is good and quite effective - the only point requiring a little of attention is the punctuation (commas are missing, sometimes, where we would expect them). 

  All in all, a good article, which surely deserves to be published, after a reasonably minor revision. 

  Thank you very much. 

Author Response

Please see the attachment. Thanks a million!!!!! 

Reviewer 2 Report

This study presents an analysis for intent recognition by deep-learning models. I think this paper requires major revision before publication. My comments are given below:

1.(Page 12)In Table 4, the F1 score is not high. Please discuss the result by using recall rate and precision rate. 

2.(Page 6)The statement for the attention layer in sub-section 4.2.3 is unclear. How can we train this layer?

3.(Page 6)The statement for Transformer in section 5 is too brief and unclear. It is hard for a reader to redo the experiments.

4.(Page 7)In Fig. 2, the output of the LSTM is a word sequence. Please explain how to bridge the self-attention layer.

5.English written needs proofread by a native English speaker.

A.(Line 21 on page 1) “attentive rnn; attentive lstm” should be revised as “attentive RNN; attentive LSTM”.

B.(Line 21 on page 1)”attention” is not adequate to be a keyword.

C.(Line 159 on page 4)”i.e.” should be revised as “i.e.,”. Line 332 on page 9 also has the same problem.

6.(Line 12 on page 1)The full name of BETO should be provided when it first appears.

7.(Page 4)The sub-grid lines in Fig. 1 should be removed.

8.(Line 225 on page 6)g does not appear in eq.(1).

9.(Page 8)Some sub-sections can be combined into one section. 

10.(Page 9)”bert” in Table 3 should be capitalized.

11(Page 11)The sub-grid lines and title should be removed in Fig. 3. The y-label “f1” should be revised as “F1”.

Author Response

Please see the attachment. Thanks a million!!!

Round 2

Reviewer 2 Report

The authors have improved the quality of this paper. I think this paper can be accepted for publication.